# Microbial Proteins in Stomach Biopsies Associated with Gastritis, Ulcer, and Gastric Cancer

**DOI:** 10.3390/molecules27175410

**Published:** 2022-08-24

**Authors:** Shahid Aziz, Faisal Rasheed, Tayyab Saeed Akhter, Rabaab Zahra, Simone König

**Affiliations:** 1Patients Diagnostic Lab, Isotope Application Division, Pakistan Institute of Nuclear Science and Technology (PINSTECH), Islamabad 44000, Pakistan; 2Department of Microbiology, Faculty of Biological Sciences, Quaid-i-Azam University, Islamabad 45320, Pakistan; 3IZKF Core Unit Proteomics, University of Münster, 48149 Münster, Germany; 4The Centre for Liver and Digestive Diseases, Holy Family Hospital, Rawalpindi 46300, Pakistan

**Keywords:** gastric cancer, gastritis, ulcer, proteomics, *Helicobacter pylori*

## Abstract

(1) Background: Gastric cancer (GC) is the fourth leading cause of cancer-related deaths worldwide. *Helicobacter pylori* infection is a major risk factor, but other microbial species may also be involved. In the context of an earlier proteomics study of serum and biopsies of patients with gastroduodenal diseases, we explored here a simplified microbiome in these biopsies (*H. pylori*, *Acinetobacter baumannii*, *Escherichia coli*, *Fusobacterium nucleatum*, *Bacteroides fragilis*) on the protein level. (2) Methods: A cohort of 75 patients was divided into groups with respect to the findings of the normal gastric mucosa (NGM) and gastroduodenal disorders such as gastritis, ulcer, and gastric cancer (GC). The *H. pylori* infection status was determined. The protein expression analysis of the biopsy samples was carried out using high-definition mass spectrometry of the tryptic digest (label-free data-independent quantification and statistical analysis). (3) Results: The total of 304 bacterial protein matches were detected based on two or more peptide hits. Significantly regulated microbial proteins like virulence factor type IV secretion system protein CagE from *H. pylori* were found with more abundance in gastritis than in GC or NGM. This finding could reflect the increased microbial involvement in mucosa inflammation in line with current hypotheses. Abundant proteins across species were heat shock proteins and elongation factors. (4) Conclusions: Next to the bulk of human proteins, a number of species-specific bacterial proteins were detected in stomach biopsies of patients with gastroduodenal diseases, some of which, like those expressed by the *cag* pathogenicity island, may provide gateways to disease prevention without antibacterial intervention in order to reduce antibiotic resistance.

## 1. Introduction

The American Cancer Society estimates about 26,380 new cases of stomach cancer and about 11,090 related deaths in the US in 2022 [1]. In the early 20th century, gastric cancer (GC) was the leading cause of cancer death in the US, a situation which is continuously improving—the number of new cases has been dropping by ~1.5% each year over the last 10 years [1]. This trend has been associated, besides the reduction in other risk factors [2], with the decline in *Helicobacter pylori* infections [1,3,4]. In other countries, particularly in East Asia, GC is still much more common as a result of insufficient sanitary standards and poor living conditions [3,4,5,6,7]. For instance, the frequency of GC in Pakistan has increased from 6.4% (study published in 2014 [8]) to 9.1% (our main study, see below, published in 2022 [9]).

*H. pylori* has been classified as a class I carcinogen by the International Agency for Research on Cancer [10]. It is the most common bacterial infection worldwide, affecting more than 50% of the global population, but only 1–3% of infected individuals ever develop GC [11,12]. Many other factors such as host constituents, environmental conditions, and the microbiota of the stomach may influence the development of gastric malignancies [12,13]. The GC microbiome has received increasing attention during the past years (for further reading of the role of non-*H. pylori* bacteria in GC development, see [14]). The stomach inactivates ingested microorganisms by exposing them to gastric fluid (containing gastric acid and proteolytic enzymes) and thus functions as a defensive barrier to harmful microbiota [15]. However, in GC patients, the stomach pH is higher and the microbiome, conclusively, changed [13,15,16]. It has been established that *H. pylori* infection can lead to achlorhydria and decreased acid secretion, thus directly contributing to alterations in the gastric microbiota (Figure 1) [11,13]. It is however not clear if the new microorganisms favor carcinogenesis or are only a result of the changed microenvironment [13].

The investigation of the GC microbiome is not straightforward and is a cause of much debate (for a recent review of pros and cons, see [13]). Arguments concern both the sample source (mostly biopsies today) and the analysis technique (16s rRNS and whole genome sequencing); the reproducibility of the results is thus limited [13]. Moreover, comparative data from the healthy stomach are hardly available for ethical reasons. Nevertheless, the carcinogenesis cascade shown in Figure 1 is widely accepted, as well as the role of microbial virulence factors (see below) [11,12,13,15,16].

While many microbial genomics data for gastric diseases are available, their complementation with protein information is still a considerable challenge [17,18]. Most studies such as the region-resolved mucosa proteome of the human stomach focused on human proteins only [19]. Metaproteomics requires the parallel measurement of thousands of proteins from an unknown number of species, namely the host and the cohabiting complex microbiome. This task presents problems on all fronts from sample handling to data mining. For instance, sample preparation methods are known to bias the recovery of host and microbial proteins, respectively [20]. In terms of the analysis of mass spectrometry (MS)-based protein expression data, databases of all organisms present are, in principle, needed, but not for every species yet available. Furthermore, the parallel use of many proteome databases extends the search space dramatically, leading to long calculation times and increased numbers of false-positive assignments, not the least due to homology effects. As a compromise, scientists study simplified microbiomes [21]. Moreover, bioinformatics approaches such as deep learning [22] and iterative searches [18] try to find a way to handle the mass of data, but they are all limited by the spectral quality of the MS raw data and the complexity of the sample.

It is, despite all limitations, of interest to learn about the presence of abundant microbial proteins in biopsies, as they may exert an influence on the tissue environment. We have thus tested MS-based comparative protein expression analysis data of gastric biopsies from patients suffering from gastritis, ulcers, as well as GC in a sub-project of a large study (for results on human proteins, see companion paper [9]). The general aim of the entire project was to generate data, which assist in reducing the number of referrals to endoscopy, which has considerably increased in recent years. This has been achieved based on MS human protein expression results, with the proposal of a lab test of the fibrinogen-to-albumin ratio at the primary to secondary care level, which recognizes gastritis. Our study [9] came with a wealth of meta-data, clinical information, and histopathological evaluation, along with *H. pylori* infection status (Aziz, S.; König, S.; Umer, M.; Iqbal, S.; Akhter, T. S.; Ahmad, T.; Zahra, R.; Ibrar, M.; Rasheed, F. Risk factor profiles for gastric cancer prediction with respect to *Helicobacter pylori*: A study of a tertiary care hospital in Pakistan. In submission). We have compared the biopsy and serum protein profiles of 219 patients and elucidated two GC serum marker panels, 29preGC-P (29 proteins significant in early-stage cancer) and 10GC-P (10 proteins significant in advanced cancer) [9].

Here, we set these results in perspective to the data obtained for the bacterial proteins detected in stomach biopsies. In the past, *H. pylori* was considered the only bacterium which can reside underneath the gastric mucosa and may be involved in gastroduodenal clinical complications and especially GC. *H. pylori* is known to have various virulence factors including vacuolating cytotoxin A (VacA) and cytotoxin-associated gene pathogenicity island (cagPAI). CagPAI encodes the type 4 secretion system (TSS4) to translocate cytotoxin-associated gene A (cagA) protein in gastric epithelial host cells to initiate carcinogenesis [23]. This concept has now changed with the emerging field of the proteomics of the human microbiome. The human gastrointestinal tract contains trillions of microbes that can live therein by forming a symbiotic relationship with their host; they play a vital role in human health and disease [24]. *Bacteriodes*, *Fusobacteria*, A*ctinobacteria*, and *Firmicutes* were, for instance, enhanced in intraepithelial neoplasia and gastric cancer, as detected by 16S rRNA sequencing [25]. This identification of other microbial species in gastric mucosa along with *H. pylori* has challenged the previous reports. We have selected the species *H. pylori*, *Acinetobacter baumannii*, *Escherichia coli*, *Fusobacterium nucleatum*, and *Bacteroides fragilis* because those organisms have been associated with gastrointestinal cancer [12,26,27,28,29]. *B. fragilis* produces toxins to cause colon cancer [30] and *F. nucleatum* was associated with poor prognosis in patients who were diagnosed with Laurens’s diffuse type GC [27]. *A. baumannii* is a multidrug-resistant bacterium [31], and *E. coli* k12 plays a role in the upregulation of programmed cell death ligand-1 (PD-L1) via the NF-κB pathway in the inflamed tissues of the intestine for immune evasion [32]. This is the first investigation, to the best of our knowledge, which provides results on the protein level rather than reporting genes associated with various gastrointestinal diseases.

## 2. Results and Discussion

### 2.1. Patients and Samples

Extensive details on patients and samples are available in the companion paper to this project [9]. Briefly, symptomatic patients having upper gastroduodenal problems (acid reflux, abdominal pain, heartburn, vomiting, bloating) attending the Center for Liver and Digestive Diseases, Holy Family Hospital, Rawalpindi, for gastroduodenal endoscopy were enrolled. Biopsies were available from 75 patients and they were divided into groups according to the gastroduodenal clinical manifestations and histopathological evaluation (normal mucosa—NGM (n = 12), mild gastritis—MiG (n = 11), moderate gastritis—MoG (n = 11), marked gastritis—MaG (n = 11), pan gastritis—PanG (n = 5), ulceration—U (gastric ulcer, duodenal ulcer, n = 12), and GC (first and advanced stage, n = 13)). Most of the study participants (70%) were *H. pylori* positive. Gastric biopsy specimens were, if possible, collected from the normal (N) and adjacent diseased (D) parts of the stomach antrum during gastroduodenal endoscopy.

### 2.2. General Results

Using the high-definition MS data sets of the tryptically digested proteins from biopsies (for experimental details and the analysis of the human proteins see companion paper [9]), the combined Uniprot protein database of *H. pylori*, *A. baumannii*, *E. coli*, *F. nucleatum*, and *B. fragilis* was searched (Table 1). A total of 304 matches were found requesting a minimum of two peptide hits. The data are given in the Supplement (Appendix A: all data; Appendix A: data for *H. pylori*-positive samples, diseased sites; Appendix A: GC vs. MaG/MoG; Appendix A: GC vs. U; Appendix A: NGM vs.: Appendix A—GC, Appendix A—MaG, Appendix A—MiG, Appendix A—MOG, Appendix A—PanG, Appendix A—U); for an overview of the whole data set, see Table 1 (Please be aware that statistical results depend on the data set studied. Thus, they may differ for the various comparisons shown in the Supplement). As is typical in proteomic analysis, for several hits in the data lists, multiple accession numbers were reported for homologous proteins, despite the fact that at least one unique tryptic peptide had to be detected. Thus, in order to clarify the true contribution of an individual protein, orthogonal analyses have to follow. Here, we discuss the results of this first global analysis.

As no enrichment of any kind has been performed, those 304 protein matches represented the most abundant proteins for each species. Chaperones, ribosomal proteins, enzymes, and elongation factors were detected as expected. The concentration of individual proteins in different organisms has been investigated by others with two-dimensional gel electrophoresis and ^14^C labeling, MS, and fluorescent light microscopy, which showed that especially these proteins are highly abundant [33]. For instance, the elongation factor EF-Tu, responsible for mediating the entrance of tRNA to the free site of the ribosome, was characterized as the most abundant protein with a copy number of ≈60,000 proteins per bacterial genome [33]. Newer work demonstrated in *E. coli* cytosol that highly concentrated proteins were involved in protein synthesis, energy metabolism, and binding, while proteins associated with transcription, transport, and cellular organization were relatively rare (dynamic range of ~10^5^) [34]. We noted that some proteins such as elongation factors 4 (EF4, *lepA*), TS (EF-TS, *tsf*), and Tu (EF-Tu, *tuf*) were found for several of the bacterial species and wondered about false-positive assignment due to homologous peptides or limited spectral information. Blast analysis in the Uniprot database revealed 76.8% homology of *H. pylori* EF4 to that of *Wolinella succinogenes*; the homology for the species we considered was in the range of 50–60%. EF4 of *B. fragilis* was 88.5% homologous to that of *Paraprevotella clara*. Prevotella are among the enriched genera proteobacteria in GC [11]. Thus, our data, and metaproteomics data generally, should be used with care; the true presence of a particular protein of a specific species always needs to be proven and validated by other analytical methods [35]. Therefore, we further analyze our results in a conservative manner considering only proteins assigned with a confidence score better than 50. Below, we discuss the hits, which were of special importance for each bacterial species, based on the global analysis containing all data sets (Table 1). Specific data for binary comparisons of individual sample sets may slightly differ from the global analysis and are available in the Supplement.

### 2.3. H. pylori

*H. pylori* uses chemotaxis to avoid areas of low pH and also neutralizes the acid in the stomach environment by producing large amounts of urease, which breaks down urea to carbon dioxide and ammonia [36]. The latter is toxic to epithelial cells as are biochemicals produced by *H. pylori* such as proteases, VacA [37], and some phospholipases [36]. *CagA* can also cause inflammation and is a potential carcinogen [38,39]. It is absent in asymptomatic human carriers [40]. TSS4 expressed by cagPAI delivers CagA and inflammation-inducing peptidoglycan to the stomach epithelial cells, starting the inflammation cascade [41]. We detected TSS4 protein CagE, but not CagA and VacA, which was possibly a matter of concentration. The *cagE* gene has been published as a biomarker for erosive gastritis [42] and as a virulence factor associated with duodenal ulceration in children [43]. On the genomic level, *cagE* was named as a prognostic marker for intestinal and diffuse gastric cancer [44]. We found its highest abundance in PanG (3.5-fold more than the overall lowest concentration found in GC, Table 1); in U vs. NGM, its concentration was not very different (fold 1.9, Appendix A). Thus, on the protein level, CagE does not appear to be a GC marker in tissue biopsies.

Primosomal protein N’ was present more than 4-fold in U vs. NMG (overall highest mean fold value, Table 1); 2.5-fold in GC vs. NMG (Appendix A). It is a member of the replication restart primosome, which is essential for bacterial survival [45]. All other proteins changed only between two- and three-fold, including DNA topoisomerase I, a protein that is crucial in maintaining DNA superhelicity, thereby regulating various cellular processes [46]. The chaperone protein HtpG was fairly balanced in all biopsies; at the maximum, it was slightly more abundant in PanG vs. GC biopsies (2.3-fold). Interestingly, the transcription of the monocistronic *hptG* gene was switched off after heat shock [47]. Successful pathogens have developed robust chaperone systems to conquer the stressful host response to infection so that increasingly the potential of exploiting pathogen chaperones as drug targets is discussed [48].

As noted above, EF4 and EF-TS were also abundant, but more so in gastritis than GC. While both proteins are involved in protein synthesis [49,50], EF4 also promotes cell death under lethal stress [51]. EF-Tu, which was suggested as a potential adherence factor of *H. pylori* during pathogenesis [52,53], was detected here at 6.3-fold more abundance in mild gastritis than in GC. Other proteins have been found in the biopsies, but with no great change between sample types (fold change < 2) such as copper-transporting ATPase. *H. pylori*-associated gastritis is accompanied by a deficiency of copper in gastric epithelial and endothelial cells [54].

### 2.4. F. nucleatum

*F. nucleatum* is an oral bacterium commensal to the human oral cavity that plays a role in periodontal plaque [55]. It is also known that *F. nucleatum* creates a pro-inflammatory environment, which is conducive to tumor growth in colon cancer [56]. We detected α1-4 glucan phosphorylase, which was seen 3.5-fold more in MaG than in GC (Table 1). This enzyme has been mainly studied in *E. coli*; roles in the regulation of endogenous glycogen metabolism in periods of starvation, stress response, or adaptation were suggested [57]. The TetR family transcriptional regulator (3.1-fold PanG > NGM) is associated with antibiotic resistance, but also with roles in metabolism and quorum sensing [58]. The majority of the detected proteins exhibited fold changes < 3 including EF-G, chaperone protein ClpB, and ATP synthase β. DNA/RNA helicase (DEAD/DEAH BOX family) belongs to the superfamily 2 RNA helicase proteins, which chaperone folding and promote rearrangements of structured RNAs [59], it appeared fairly balanced across the biopsies of this project (2.1-fold).

### 2.5. E. coli

*E. coli* and other facultative anaerobes constitute about 0.1% of gut microbiota [60]. Most *E. coli* strains are harmless, but some serotypes can cause serious food poisoning in their hosts. The harmless strains are part of the normal microbiota of the gut and can benefit their hosts by producing vitamin K_2_ and preventing the colonization of the intestine with pathogenic bacteria [61,62]. More than a hundred complete genomic sequences of *Escherichia* are known and they show a remarkable amount of diversity; around 80% of each genome can vary among isolates [63]. Each individual genome contains between 4000 and 5500 genes (38% identified experimentally [64]), but the total number of different genes among all of the sequenced *E. coli* strains (the pangenome) exceeds 16,000. It has been assumed that two-thirds of the *E. coli* pangenome originated in other species and arrived through the process of horizontal gene transfer [65]. The transfer of DNA from one bacterial cell to another through the intervening medium appears to be part of an adaptation to stressful conditions, including those that cause DNA damage [66]. We detected DNA mismatch repair protein MutS [67] and Rpn family recombination-promoting nuclease/putative transposase [68], which are involved in these functions.

Besides the chaperone DnaK (maximum fold change 4.6, PanG vs. NGM), lipid-A-disaccharide synthase, which is a virulence factor in pathogenic bacteria [59], exhibited considerable differences across the biopsies; it was five-fold more abundant in MoG than in GC. All other proteins only changed less than three-fold and they were upregulated in MaG and U. The outer membrane usher protein YehB is part of the chaperone/usher pathway, which assembles adhesive and virulence-associated pili [69]. Multifunctional conjugation protein TraI has a role in antibiotic resistance propagation [70]. Sulfoquinovosidase is the gateway enzyme to sulfoglycolytic pathways [71].

### 2.6. B. fragilis

*B. fragilis* is part of the normal microbiota of the human colon and is generally commensal [72]. It is essential for healthy gastrointestinal functions such as mucosal immunity and host nutrition [73]. *B. fragilis* is an aerotolerant, anaerobic chemoorganotroph capable of degrading a wide variety of glycans, biopolymers, polysaccharides, and glycoproteins. It produces fatty acid as a byproduct of carbohydrate fermentation which can then be used as energy by the host; it was shown that animals lacking gut bacteria require 30% more caloric intake to maintain body mass [74]. The *B. fragilis* group is the most commonly isolated Bacteroidaceae in anaerobic infections, and *Bacteroides* are characterized by the highest resistance rates amongst anaerobic bacteria [75]. In biopsies, major changes were indicated for ParB-like partition protein (5.6-fold MoG > NGM). In bacteria, partition systems contribute to the faithful segregation of both chromosomes and low-copy-number plasmids [76]. Besides EF-Tu (3.5-fold PanG > GC), β-mannosidase was higher regulated than 3-fold (3.8-fold MaG > GC). It plays an important role in the polysaccharide degradation pathway [77]. Other proteins specifically seen in this organism were a putative cobalamin biosynthesis-related membrane protein [78], type I restriction enzyme R protein [79], 4-α-glucanotransferase, and metal-dependent hydrolase [80].

### 2.7. A. baumannii

*A. baumannii* can be an opportunistic pathogen in humans, affecting people with compromised immune systems, and is becoming increasingly important as a hospital-derived infection [81]. *A. baumannii* has also been identified as an ESKAPE pathogen, a group of pathogens with a high rate of antibiotic resistance that are responsible for the majority of nosocomial infections [82]. It can survive on artificial surfaces for an extended period of time, likely due to its ability to form biofilms [83]. In our experiments, regulated more than four-fold (PanG vs. MiG) was a tape measure domain protein, which received this name because the length of the corresponding gene is proportional to the length of the tail of a bacteria-infecting phage [84]. In fact, as the nosocomial pathogen *A. baumannii* is multidrug-resistant, phage therapy as an alternative to antibiotic therapy has been recovered [85]. Interestingly, there was also a phage integrase family protein detected predominantly in PanG biopsy samples (two-fold >NGM), a protein group which has risen to prominence as genetic tools [86]. Translation initiation factor IF-2 (2.4-fold PanG > NGM) synthesis was shown to be induced following cold stress, while ribosome synthesis and assembly slowed down [87]. Upregulated in U was a short-chain dehydrogenase family protein (3.9-fold vs. MiG). Its structure has been described from a genomic island of a clinical strain of *A. baumannii* [88]. Additionally, abundant in U was the pyruvate dehydrogenase E1 component (3.1-fold > NGM). It is part of the pyruvate dehydrogenase complex enzymatic system that is crucial in cellular metabolism as it controls the entry of carbon into the Krebs cycle [89].

### 2.8. Limitations

As discussed already in part above, this study suffers from limitations. We have been fortunate in having available to us samples of different gastrointestinal diseases for this project, but for some subgroups such as PanG, only a few biopsies were available, reducing the power of the analysis. The results should thus be considered with care in their context. Moreover, while the proteomic analysis of total proteomes has great value in providing an overview of the sample composition, it is a general and, in a way, a superficial experiment, which is not always able to properly identify homologous proteins or isoforms. This was discussed for elongation factors in Section 2.2. Subsequent orthogonal analyses need to be employed to pinpoint the presence of proteins of interest. Furthermore, proteomic experimental results are big data sets containing tens of thousands of peptide signals. Their analysis requires software support, which can generate false-positive matches depending on the search and filter criteria chosen. The best way to exclude these is still the manual control of individual mass spectra, which is not feasible for large data sets [35]. We have thus used both the number of detected peptides and the confidence score to eliminate false hits as much as possible.

## 3. Conclusions

No such comprehensive proteomics research study has been conducted before, which comprised various gastroduodenal afflictions with respect to *H. pylori* infection. This is also the first investigation to the best of our knowledge, which has provided results on the protein level rather than reporting regulated genes. In the past, *H. pylori* was considered the only bacterium living beneath the gastric mucosa and which may be involved in gastric cancer initiation. We have here detected proteins of four other bacterial pathogens, *A. baumannii*, *E. coli*, *F. nucleatum*, and *B. fragilis*, in biopsy specimens of gastric diseased patients.

Our measurements did not paint a comprehensive picture of the microbial proteins; they rather gave a first impression as we focused on the most confident protein matches. Ambiguity remained with regard to isoforms and homologous proteins, and individual hits will need to be validated. Abundant bacterial proteins were assigned, such as elongation and heat shock factors, which seemed to be highly concentrated across species. Other proteins, such as the *H. pylori* virulence factor CagE [42,43], appeared to be specific. For the oral bacterium *F. nucleatum*, we detected α1-4 glucan phosphorylase, which is involved in the regulation of endogenous glycogen metabolism during stress [57]. In *E. coli*, the virulence factor lipid-A-disaccharide synthase [59] was found especially in gastritis. So was β-mannosidase in the commensal bacterium *B. fragilis*; the enzyme is important in the polysaccharide degradation pathway [77]. In the opportunistic pathogen *A. baumannii*, the pyruvate dehydrogenase E1 component was regulated, which is crucial in cellular metabolism [89]. Interestingly, bacterial proteins of interest (Table 1) were more abundant in advanced gastritis than in NGM and GC. This finding could be reflective of increased microbial involvement in mucosa inflammation in line with current hypotheses [90]. For the *H. pylori* virulence factor CagE, evidence to that effect already exists [42,43].

The eradication of *H. pylori* has been considered as a measure to prevent GC [82,83] because it was recognized as a major risk factor. Antibiotic treatment obviously affects microbiota in general, however, leading to the temporary dysregulation of the microbiome and supporting antibacterial resistance. Thus, such harsh treatment needs to be applied with care and more selective; personalized approaches are called for (for more information on the exploitation of microbiota in personalized medicine, see [91]). Clearly, in disease research, the entire microbiome in its complexity is of importance. The findings of this study are helpful in redirecting the research focus on bacterial pathogens other than *H. pylori* residing underneath the gastric mucosa at acidic pH. The lack of diagnosis of these pathogens may be involved in the increase in antimicrobial resistance when treating *H. pylori* infection. It may thus be necessary to test for more bacterial species in routine lab work.

Moreover, information on the presence and regulation of various proteins of bacterial pathogens assists the scientific community in its investigation of microbial species in the context of gastric disease treatment and prevention such as the design of novel inhibitors to halt the spread of bacterial infections and GC. More research is still needed on the molecular pathways and host–pathogen interactions in gastric disease progression.

## 4. Materials and Methods

### 4.1. Permissions

Ethical approvals were obtained from the Ethical Technical Committee of the Pakistan Institute of Nuclear Science and Technology (PINSTECH), Islamabad (Ref.-No. PINST/DC-26/2017), the Bioethics Committee of Quaid-i-Azam University, Islamabad (Ref.-No. BBC-FBS-QAU2019-159), the Institutional Research Forum of Rawalpindi Medical University, Rawalpindi (Ref.-No. R-40/RMU), and the Ethics Committee of the University of Münster, Germany (Ref.-No. 2021-339-f-N). Informed written consent was obtained from each participant.

### 4.2. Protein Expression Analysis

Sample preparation and measurement were described in reference [9]. Briefly, proteins were extracted from the biopsies, tryptically digested, and subjected to reversed-phase liquid chromatography coupled to high-definition MS with Synapt G2 Si/M-Class nanoUPLC (Waters Corp., Manchester, UK). Label-free data-independent quantification experiments were analyzed with Progenesis for Proteomics (QIP, Nonlinear Diagnostics/Waters Corp.) using the combined Uniprot databases for from *H. pylori*, *A. baumannii*, *E. coli K12*, *F. nucleatum and B. fragilis* (download 8/25/2021). At least two peptides were required for protein assignment. Fold values as well as ANOVA *p*-values and confidence levels were also determined using Progenesis.

## Figures and Tables

**Figure 1 molecules-27-05410-f001:**
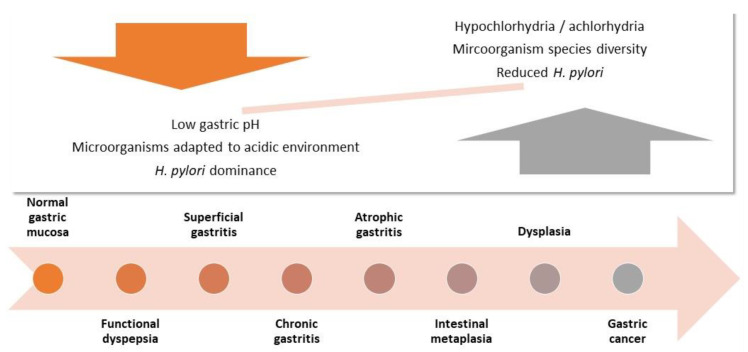
Changes in gastric microbiota and microenvironment during carcinogenesis (adapted from [11,13,16]). Following infection, *H. pylori* dominates the gastric microbiota leading to chronic gastritis, reduced acid secretion, and alterations in the gastric microbiome.

**Table 1 molecules-27-05410-t001:** Significantly changed bacterial proteins detected in stomach biopsies in the global analysis of all samples (Anova *p* < 0.02, confidence score > 50, fold change > 2, excluding proteins of no known function). For each MS data set, which was presented to Progenesis software, the cumulated ion intensities were used to calculate fold values for protein matches. The groups showing the highest and the lowest mean values are reported. In the case of multiple assignments of the same protein, only the best match is given. For details see Appendix A, for binary comparison Appendix A. NGM has been italicized, and GC was marked in bold. Interestingly, the lowest mean condition is populated by NGM, MiG, and GC indicating more involvement of bacterial proteins in gastritis and ulcer.

Accession	Peptide Count	Unique Peptides	Confidence Score	Max Fold Change	Highest Mean Condition	Lowest Mean Condition	Mass	Description	Name
	** *H. pylori* **								
Q48252	32	30	201	3.5	PanG	**GC**	112,648	Type IV secretion system protein CagE	cagE
P55991	30	10	195	2.4	MaG	MiG	84,222	DNA topoisomerase 1	topA
Q9ZMM2	11	2	84	2.3	PanG	**GC**	71,311	Chaperone protein HtpG	htpG
B2USI5	13	4	77	2.8	MoG	**GC**	67,412	Elongation factor 4	lepA
Q9ZKE4	12	3	74	4.4	U	*NGM*	71,164	Primosomal protein N’	priA
B5Z9I6	11	3	64	2.7	PanG	*NGM*	39,970	Elongation factor Ts	tsf
Q9ZKF6	8	7	52	2.3	MaG	MiG	62,699	30S ribosomal protein S1	rpsA
	** *F. nucleatum* **								
Q8RGE2	46	40	372	2.8	MaG	NGM	50,219	ATP synthase subunit beta	atpD
Q8RF61	18	14	109	3.5	MaG	**GC**	91,782	Alpha-1_4 glucan phosphorylase	FN0857
Q8R5Z3	13	12	85	2.1	PanG	*NGM*	11,0809	DNA/RNA helicase (DEAD/DEAH BOX family)	FN1974
Q8RFA3	10	8	75	3.1	PanG	*NGM*	44,947	Transcriptional regulator_TetR family	FN0813
Q8RHQ8	12	8	75	2.9	PanG	**GC**	97,405	Chaperone protein ClpB	clpB
Q8RIF1	10	8	73	2.3	PanG	MiG	30,287	Oligopeptide transport ATP-binding protein oppD	FN1649
Q8R604	11	8	60	2.1	PanG	MiG	78,086	Protein translation elongation factor G (EF-G)	FN1546
	** *E. coli* **								
P33341	17	15	99	2.9	MoG	**GC**	92,510	Outer membrane usher protein YehB	yehB
P23909	17	3	99	2.7	PanG	*NGM*	95,589	DNA mismatch repair protein MutS	mutS
P32138	13	1	79	2.2	MaG	**GC**	77,902	Sulfoquinovosidase	yihQ
A0A6D2W465	14	13	75	4.6	PanG	*NGM*	69,172	Chaperone protein DnaK	dnaK
P14565	10	9	66	2.7	MaG	*NGM*	19,2016	Multifunctional conjugation protein TraI	traI
A0A6D2XSN2	12	9	65	2.1	U	MiG	36,025	Rpn family recombination-promoting nuclease/putative transposase	FAZ83_12895
A0A4S5B3J9	11	10	62	5.0	MoG	**GC**	42,787	Lipid-A-disaccharide synthase	lpxB
A0A6D2XN30	8	7	58	2.5	PanG	MiG	12,295	50S ribosomal protein L7/L12	rplL
	** *B. fragilis* **								
A0A380YYU9	77	69	452	2.5	MaG	**GC**	55,261	ATP synthase subunit beta	atpD
Q5LEE2	31	29	182	3.8	MaG	**GC**	99,598	Beta-mannosidase	bmnA
A0A380YZS6	27	25	166	3.0	PanG	**GC**	148,490	Putative cobalamin biosynthesis-related membrane protein	cobN_1
Q5LEB7	23	20	143	2.6	MiG	**GC**	107,262	Type I restriction enzyme R Protein	hsdR
Q5L9C7	19	16	113	2.1	PanG	**GC**	106,621	4-alpha-glucanotransferase	malQ
Q5L8N9	10	7	57	5.6	MoG	*NGM*	73,150	ParB-like partition proteins	parB_1
A0A380YS49	11	9	56	3.5	PanG	**GC**	43,808	Elongation factor Tu	tuf
Q5L7x1	8	6	50	2.5	MoG	*NGM*	69,190	Metal-dependent hydrolase	BF9343_4047
	** *A. baumannii* **								
A0A009HW70	29	25	152	4.1	PanG	MiG	144,320	Tape measure domain protein	J512_0646
A0A009IJB7	20	16	121	2.0	PanG	*NGM*	127,978	Phage integrase family protein	J512_2803
A0A009IU53	16	9	88	2.4	PanG	*NGM*	97,429	Translation initiation factor IF-2	infB
A0A009ILQ7	12	10	66	2.9	PanG	**GC**	123,032	DEAD/DEAH box helicase family protein	J512_2067
A0A009HXA1	12	8	61	3.1	U	*NGM*	102,148	Pyruvate dehydrogenase E1 component	aceE
A0A009IT43	7	6	50	3.9	U	MiG	27,161	Short chain dehydrogenase family protein	J512_0848

## Data Availability

All data are available in the Appendix A.

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
