# Peer review of "Microbial Proteins in Stomach Biopsies Associated with Gastritis, Ulcer, and Gastric Cancer"

_molecules, 2022, doi:10.3390/molecules27175410_

Round 1
Reviewer 1 Report
The work by Aziz et al. concerns a very important topic and it is of great relevance in the context of gastric cancer pathology. A deep proteome characterization of bacteria associated to the most important gastric diseases can offer crucial information to human genomics and proteomics data and contribute to get better insight GC.
There are some points that need to be addressed before publication.
The §Abstract cites “three groups” (line 21), but more than 3 groups were analysed.
It is written that 304 bacterial proteins met the filtered criteria (line 24), these filter criteria should be defined in §Materials and methods. Similarly, when authors cite “Species-specific proteins were detected mostly with more abundance in gastritis than in GC” (lines 24-25), §Materials and methods they need to specify how they calculated the abundance (did they use a label free quantification? No quantitative comparative data are reported in Supplementary table 1).
When they write 304 proteins (line 24), do they mean “distinct proteins” ? (they are redundant, please provide the number of distinct proteins, that should be around 270).
In lines 28-29, the sentence “validation through 28 orthogonal methods and functional analysis” is not clear. The authors should conclude about the most important data of their work.
§Introduction
Line 73: the authors should specify “microbial” genomics data.
Line 102: they should define preGC-P and GC-P.
§ Results and Discussion
Line 109: Reference 21 cannot be cited because it is in submission.
§2.1 the part concerning patients and samples should be placed in §Materials section.
What about GC ? Was it intestinal or diffuse ?
§2.2, line 128: “They matched expectations in that chaperones, ribosomal proteins, enzymes and elongation factors 128 were predominantly detected” the authors should indicate how they got those results, define “predominantly”.
Line 147-150: when they cite the confidence score >50, did they use the scores reported in Supplementary table 1 ? Values are not the same.
The authors should better show their most important results, especially those discussed in details in the §Results and Discussion. It is hard to look for the cited proteins in the Supplementary Excel file.
For instance, when they cite the “primosomal protein N” as present more than 4-fold in U vs. NMG, I am going to “Sheet 9_NGM_U_HPpD”, I can find the protein in the line 34 but I cannot find the highest mean fold value cited. Since they write U vs. NMG the fold change values should be calculated as described in the text, that is to say U vs. NMG.
Authors should specify in each sheet how they got the FC values and evidence protein identifications passing the statistics (Anova p value <0.05??). The fill color of the cell should be justified. Otherwise, cells should be without color.
I think all the statistically significant proteins should be presented in the main text; the complete MS data can be shown as supplementary.
Authors should also describe the statistics applied to MS data.
Indeed, it is not clear how the authors linked results shown in the Supplementary Table with those shown in Table 1.
In “S8_NGM_PanG_HPpD”, I have found the “Type IV secretion system protein CagE OS=Helicobacter pylori” in line 8, with the characteristics shown in Table 1 (peptide count: 32; unique peptides: 30; confidence score: 201) but I cannot find the max fold change 3.5 and the reported ANOVA is 0.062.
This part and the statistics applied should be better described. Since the authors used many different diseases and it can be hard to explain the comparisons done, they may also show a schematic illustration to better describe the approach and the matches among diseases.
In the Suppl table 1, authors should check protein entries. Some of them are obsolete [for instance https://www.uniprot.org/uniprotkb/A0A237L7M0/history] and should be up-dated.
The authors should explain the meaning of “Highest mean condition” and “Lowest mean condition”.
An interesting point the authors did not afford is to discuss proteins shared by the bacteria.
Moreover, it is hard to understand the criteria by which bacterial proteins are listed in Table 1.
Maybe the authors can divide it into 5 parts with subheadings reporting the bacterial species?
In §Materials, even if authors used the same patients as them used in [11,21], they should even briefly show some data concerning them. Did they use all biopsies and all serum? Maybe they may simply show the total number of samples per disease, to have an idea about the entity of the study.
To get better insight the kind of gastrointestinal disease and the reason of their global analysis in the context of GC characterization, I may propose the authors to briefly describe the diseases analysed.
How did the diagnose mild gastritis or moderate gastritis, marked gastritis..etc pan gastritis, PanG is a pan gastritis with ulceration?
Why serum samples (n=75) were more numerous then biopsies one (n=219) ?
§References
The authors should check the bibliographic entries. References must be numbered in order of appearance in the text (including table captions and figure legends) and listed individually at the end of the manuscript.
96 references is quite too much for a research paper.
Authors should decrease this number.
§Title
Finally, it is not evident the reason of the title. I suggest the authors to change it into something more general and better representing the whole work.
Reviewer 2 Report
Thank you very much for the opportunity to review the manuscript entitled “No correlation of the presence of Helicobacter pylori type IV secretion system protein CagE with gastric cancer in stomach biopsies. Although the manuscript has valuable evidence, many questions remain to be addressed in the next experiments. I have some comments which should be considered as below:
Major comments:
In the introduction section, further explanations and corrections are required for a better understanding of the interaction between microbial species, and gastric cancer. In the current form, it would be difficult to generate a comprehensive insight into their interactions, however, a better organization and structure of the different issues can help to incorporate more relevant information into the topic. Also, more previous experimental evidence should be needed to discuss and support the main question of the study and the references should be updated.
It is better to describe the main gaps in this field and novelty of the study and what can be worth it for researchers in this field in the introduction section.
Similar to the introduction section, in the discussion section, the given evidence about the interaction between the individual ideas is very weak. Also, the novelty of the study should be addressed significantly. Further explanations and drafting are required to highlight the novelty of the study.
Another drawback of this article is that it lacks comments about the clinical significance and future perspectives of the finding.
Please include a limitation statement in the manuscript. Although the manuscript has valuable evidence, many questions remain to be addressed in the next experiments.
Minor comments:
The full term for which an abbreviation stands should precede its first use in the text.
The manuscript should contain all detailed information about the experiments. All your methods should have the appropriate citation of references. Check and revise them properly
Some references are extremely out of date and must be replaced by more recent and relevant original studies.
Some sentences are too long and difficult to follow. Please rephrase and divide it into two sentences.
Although the language of the manuscript is adequate and easy to follow, some sentences would benefit from editing.
Reviewer 3 Report
Aziz et al studied the correlation between Helicobacter pylori type IV secretion system protein CagE and gastric cancer in stomach biopsies using proteomics data. The authors identified specific bacterial species from other literature as their target microorganisms that contribute to gastric cancer related disease prognosis. The overall study design is poor, lacks hypothesis and clear experimental design, and the bold conclusion made by the authors using bioinformatic analyses put them against the already established scientific evidence that CagE is indeed associated with gastric cancer (see https://doi.org/10.1073/pnas.1903798116, https://doi.org/10.1016/j.ijid.2009.09.006, and https://doi.org/10.1016/j.meegid.2020.104477). Bioinformatic analyses have their limitations and, therefore, making such bold conclusion based only on bioinformatic analyses is not expected in scientific literature.
To make the study more relevant, the authors could present the study as bioinformatic/proteomic analysis of microbiome in stomach biopsies associated with gastric cancer. This would provide some valuable information.
Abstract: While the abstract is informative, it appears that the authors have just threw some information there without making it coherent. Change the writing style to more active form for better readership.
Line 42: Include the year of data collection information. Stating the increase without providing year of data collection information does not make any sense.
Line 49: Remove “on the one hand”
Line 52: Remove “on the other hand”…Include…However, in GC patients, the stomach pH…
Line 43-60: The authors included references for statements that does not include what information the authors are referring to. It’s not a reviewer’s/reader’s job to search for relevant information in citations the authors are referring to. References are rather used to support information that is already presented. This whole paragraph needs to be formatted for better presentation and readership.
Line 49 and 58: Remove the review references...include specific study that presented relevant information
Line 90-95: Split the sentence into two separate sentences for better readership
Line 101: Replace “had” with “have”
Line 109: What details are the authors referring to? Just including the references is not enough where patients and samples information is a critical part of this study. Present the data in a Table within the text so correlations can be made with other metadata and proteomics datasets.
Round 2
Reviewer 1 Report
The new version is certainly better and can be accepted.
Reviewer 2 Report
Dear Editor
I think the authors have successfully revised their manuscript entitled “Microbial proteins in stomach biopsies associated with gastritis, ulcer, and gastric cancer” and this paper is mostly easy to follow. In the revised version of the manuscript, the authors carefully addressed my earlier concerns points accordingly. This version of the manuscript now seems to be well-written and should provide valuable findings. Thus I recommend the paper be published.
Reviewer 3 Report
None